# Olive Stones as Filler for Polymer-Based Composites: A Review

**DOI:** 10.3390/ma14040845

**Published:** 2021-02-10

**Authors:** Sara Valvez, Alberto Maceiras, Paulo Santos, Paulo N. B. Reis

**Affiliations:** Centre for Mechanical and Aerospace Science and Technologies (C-MAST-UBI), Universidade da Beira Interior, Rua Marquês d’Ávila e Bolama, 6201-001 Covilhã, Portugal; sara.valvez@ubi.pt (S.V.); alberto.maceiras@ubi.pt (A.M.); paulo.sergio.santos@ubi.pt (P.S.)

**Keywords:** olive stone, filler, polymer, composite

## Abstract

Olives’ consumption produces copious agricultural byproducts that have accompanied humanity for millennia, but the increasing worldwide production complicates its management. Most wastes are generated during olive oil production in form of olive stones and other lignocellulosic derivatives. Industrial processes of chemical or physical nature to recover economically compounds from biomass residues are costly, difficult, and non-environmentally friendly. Cellulose, hemicellulose, and lignin biopolymers are the principal components of olive stones, which present interesting qualities as lignocellulosic fillers in polymeric composites. This review will summarize examples of composites based on thermoplastic polymers, such as polystyrene (PS), polylactide (PLA), polyvinyl chloride (PVC), polypropylene (PP), and polycaprolactone (PCL); thermosetting resins (phenol-formaldehyde, unsaturated polyesters, and epoxy) and acrylonitrile butadiene rubber/devulcanized waste rubber (NBR/DWR) blends focusing on the fabrication procedures, characterization, and possible applications. Finally, thanks to the wide disparity in polymer matrix types, the variability in applications is important, from adsorption to mechanical enhancement, showing the easiness and benefit of olive stone integration in many materials.

## 1. Introduction

The olive tree is considered one of the first trees cultivated in human history, with implications for religion (Greek mythology and the Old Testament) and social life of our ancestors. For example, the olive plant has been a symbol of peace and friendship since Ancient Greece, when the Olympic Games winners were honored with a garland of olive branches [1].

Although more than 30 olive trees species are known, only the *Olea europea L*. is edible. The modern olive tree is believed to have appeared about 5000 years ago in Mesopotamia. The Phoenicians were responsible for their initial spreading along the Mediterranean coast and, later, the Greeks and Roman Empire. The Romans used olive oil in many applications, and the olive oil trade was a major commercial success that increased the trade routes. During the following centuries, the olive-derived industry continued being increasingly important up to our day [2].

The basic oil obtention procedure consisted of squeezing olives in stone mortars and collecting the olive paste in pots. Then, with the addition of hot water, the oil was easily separated from the solid mixture, because it remained on the surface due to its lower density in relation to water. Later, the methodology was improved with the use of different technologies. The improvement in the rushing and application of pressure with new devices pressing systems, such as millstone crushers for the crushing, presses, and hydraulic enhanced the oil extraction step before phase separation, in the so-called traditional milling method. Finally, in the modern three-phase oil removal system, industrial decanters based on centrifugation, a heating process (malaxation step), and a further residue oil purification with hexane or other reagents can be used [3].

The growth of olive tree plantations, the industrialization of agriculture, and the extraction procedures have significantly increased the production of olives worldwide in recent decades. The European Union (EU) is the main producer, consumer, and exporter of olive oil, accounting for 80% and consuming 70% of the world’s production. Although the production of the EU leads the world market, production in other regions is rapidly rising. Only nine member countries (Spain, Italy, Greece, Portugal, France, Slovenia, Croatia, Cyprus, and Malta) are responsible for about 5 million hectares of olive cultivation, ~3.3 million olive growers, and an average production of ~16 million metric tons, the majority for olive oil production. As shown in Figure 1, more than 50% of the total farmed area and 63% of the whole EU production is located in Spain [4].

With the evolution of technology over time, the efficiency of production methods has increased significantly and, consequently, the production of waste has become increasingly expressive. According to the literature, the main byproducts of olive processing are: (a) Solid: Pomace (crushed stones and detritus of the pulp), twigs and leaves, (b) liquid: Fine olive pulp, unrecoverable oil, olive juice and water, named olive mill wastewater (OMWW), and (c) gaseous: Fumes generated in the malaxation step and other machinery involved processes [5]. Therefore, in addition to the olive grove being responsible for the production of a large amount of organic waste, it is also clear that the transformation process of olives is responsible for several types of waste, many of them without clear and viable economic recovery. On the other hand, waste handling is an important problem due to the high associated cost.

It is estimated that, with the most modern technology available, one ton of olives is responsible for 0.6 ton of olive mill solid residue. Many of these residues are used in the production of electrical and/or thermal energy, because the costs associated with the dispersed location of small/medium farms/properties or any advanced technological solution that implies complex chemical treatments become prohibitive for alternative solutions. Considering that reducing the cost of technology or transforming waste into value-added materials are difficult options, physical and chemical processes have been studied, especially at level of olive mill wastewater (OMWW) [6,7]. Moreover, the complex and chemically stable nature of the lignocellulosic biomass makes it difficult to separate and purify it into simpler and commercially profitable chemical compounds at low costs.

In order to overcome this problem, literature also reports studies in which lignocellulosic materials derived from olive residues are used as fillers for polymer-based composites, because biocomposites have been gaining more and more approval since the oil crisis. This proposed solution is similar to that used in composites, where synthetic fibers are replaced by natural fibers due to their biodegradability, abundance, low density, low cost, and they are non-abrasive [8,9,10,11,12]. Compared to synthetic fibers, natural fibers generally have a specific weight, around half the weight of glass fibers, and a tensile modulus very similar to that of aramid fibers [13], but it should be noted that their mechanical properties depend on cellulose content, degree of cellulose polymerization, and microfibril angle [14] as a consequence of their cultivation (variety, climate, harvest, maturity, retting degree), processing (decortication, disintegration), and fiber modification (textile and technical processes) [9,15].

Regardless of the benefits previously reported, the fact that forests are decreasing, especially in developed countries, leads to the use of other agricultural byproducts or agro-waste to reinforce polymer-based composites [16,17,18]. Therefore, in this context, this manuscript intends to review the polymeric composites based on olive stone fillers and involving different types of polymers (thermoplastic polymers, thermosetting resins, and rubber materials). For this purpose, the bibliographic research that supports the present study focused on papers published in English, on scientific documents indexed to the Scopus database, combining “olive stones and composites” or “olive pit and composites”, and only papers related to mechanical performance, or those that provide sufficient information for their estimation, were considered. Finally, in addition to a section dedicated to olives and lignocellulosic materials, the document is structured by type of polymer (thermoplastic polymers, thermosetting resins, and rubber materials) and, independently of the number of papers published, in each of these sections there are subsections dedicated specifically to a polymer, in which the studies are listed by year of publication.

## 2. Olive Waste Management, Olive Fruits, and Lignocellulosic Materials

### 2.1. Olive Waste Management and Economic Perspective

Olive mill wastewater (OMWW) resulting from the oil extraction process is the main waste, and it is estimated that values around 5.4 × 10^6^ m^3^ are produced annually worldwide. In addition to the large quantities of OMWW produced, its high-polluting power due to high biochemical oxygen demand (BOD), chemical oxygen demand (COD), total solids, organic carbon, and slightly acidic character of OMWW promotes a severe negative impact on the soil, agriculture, water quality, environment, and public health [19,20,21]. In fact, these residues are a serious environmental problem, because they have high concentrations of phenol, lipids, and organic acids, in addition to being resistant to biodegradation [22,23]. Therefore, for economic reasons, OMWW was often concentrated in evaporation ponds and left to dry over the summer, however, this method of disposal is currently prohibited in most countries. In this context, the literature reports several treatment alternatives, such as: Chemical, mechanical, physical, biological, and thermal methods, or their combinations. The most common are: (a) Chemical flocculation/coagulation, (b) land application, (c) Evaporation-Hydrolysis-Oxidation (EHO), (d) membrane processes, (e) electrolysis, (f) vacuum evaporation, (g) natural evaporation, (h) anaerobic treatment, and (i) pretreatment options [6,7]. The main technologies for OMWW treatment were summarized by Dutournié et al. [24] and reviewed by Mantzavinos and Kalogerakis [25], while Azber et al. [26] analyzed the costs for the different treatments and disposal alternatives. According to Azber et al. [26] natural evaporation, both in forced or improved forms, is the cheapest solution, while biological treatments require expensive pretreatments, careful management, and specialized work. More recently, other management options have been proposed for treatment and valorization of OMW, which aim to reduce the phytotoxicity to reuse it for agricultural purposes. However, few techniques have been applied on an industrial scale due to the high cost of the treatment plant [22]. For example, the investment cost can vary enormously, from 42,000€ in forced natural evaporation to 500,000€ in mechanical-biological pretreatment plants involving, in this case, operating costs per cubic meter of wastewater (€ × m^−3^) and calculated costs per ton of olive oil (€ × ton^−1^) around 0.05–6.82€ × m^−3^ and 1.5–67.5€ × ton^−1^, respectively [26].

The olive stone, resulting from the olive oil extraction and pitted table olive industries, is another residue with high environmental and economic interest. It is probably the residue produced in the olive fruit industrial sector with the greatest commercial interest due to its chemical/physical properties and combustion heat. In this context, because the olive stone is a lignocellulosic material (with hemicellulose, cellulose, and lignin as the main components), the literature reports several studies with methods of recovering lignocellulosic material or biomass for production of solid, liquid, or gas biofuel. In addition, other compounds with high added value can be obtained for other uses, which are synthesized in Table 1.

Regarding the possible applications mentioned above, the present manuscript will focus mainly on composite materials reinforced with olive stones, as bio-filler, in order to report its influence on the mechanical performance of such materials. Basically, in addition to intending to add value to a renewable bio-product from an economic point of view, it also aims to minimize the negative effects on the environment of certain polymers by promoting clean technologies and recycled products. For example, products such as panels, sandwich panels, tubes, or profiles, among others, can be produced by extrusion or injection technologies [27,28]. However, knowing even more the advantages that this bio-filler offers, mainly in terms of mechanical behavior, we are contributing to a wider application in the industrial sector. Therefore, it is in this context that this review intends to awaken in the reader a critical analysis of the benefits of using olive stones as reinforcement of composite materials.

### 2.2. Olive Fruit Parts

The olive fruit as a structure can be divided into three parts: (a) The skin also labeled as epicarp, (b) the pulp, flesh, or mesocarp, and (c) the stone or endocarp (Figure 2). The skin is responsible for the color, because it contains chlorophyll, anthocyanins, and carotenoids, and represents 1–3% of the weight. The pulp is the main section of the olive, representing 70–80% of the weight, and most of all components. Finally, the stone is composed of the seed and wood shell (stone) [29].

The olive stone without seed can be recovered by filtration of solid waste from the olive oil industry, while the whole olive stone (stone and seed) is obtained by separation of the pulp in the pitted table olive industry. According to Rodriguez et al. [27], the 2005/2006 world harvest alone was responsible for 2.58 and 1.73 million tons of olive stones obtained from the olive oil and table olives industries, respectively. Therefore, these values evidence the commercial interest reported in the literature by this residue produced in the olive fruit industrial sector.

### 2.3. Olive Stones’ Chemical Composition

Olives are a lignocellulosic biomass of varying chemical composition, but essentially formed by lignin, cellulose, and hemicellulose, in addition to proteins and fats (Figure 3). For example, in an average olive seed, there is 22% to 27% by weight of oil, which usually disappears during the drying process. In addition, a small percentage of C_23_–C_33_ alkanes, polyunsaturated acids, sugars, polysaccharides, and amino acids can be detected [27]. A list of the main chemical components and molecules extractable and detected in chloroform–ethanol (CHCl_3_–EtOH) is showed in Table 2.

In addition to its main use being as biomass and fuel, olive stones are useful thanks to their physical properties and useful chemical components [30]. The three main components of lignocellulosic olives are interesting enough to be separated by fractionation. Nonetheless, the fractionation of the three components is limited by chemical and physical impediments. Difficult, inefficient, and non-environmentally friendly processes are necessary to carry out the process, such as steam explosion or chemical extraction. Applications for olive stones where the excess biomass can be used without the need for extensive and complicated procedures are highly appreciated [31].

Natural cellulose is a renewable natural polymer that exists in all algae and plants, and in some marine creatures and microorganisms that can synthesize the biopolymer. Cellulose is the most common macromolecules in the world, because it represents 30 to 40% of the wood content or 65% in bast plants, which represents a global production of ≈1.5 × 10^12^ tons per year. Cellulose can appear in plants in pure form, but is usually accompanied by hemicellulose and lignin. Depending on the type of plant, the amount of cellulose varies enormously, for example, in jute (45–60 wt.%) and bamboo (40–55 wt.%).

Chemically, cellulose presents a multi-level complex structure, and several types of cellulose are known, from Cellulose I to IV. Cellulose I is the natural cellulose, cellulose II is regenerated cellulose, and III and IV can be obtained by chemical treatments [32]. (C_6_H_10_O_5_)_n_ is the chemical formula, *n* is the number of glucoses and the degree of polymerization (chain length), and many of the cellulose properties depend on that length. Furthermore, the number of glucose units varies according to the species of plant, from 300 to 10,000 units. Cellulose is formed by bundles of nano-fibrils, each of which is formed by 60% of crystallites and the rest by random amorphous domains. Each cellulose chain has many crystallites and disordered amorphous parts linked together by 1,4-β-glycosidic bonds. D-glucopyranose is the repetitive unit in this stereo-regular and linear polysaccharide.

Hemicellulose is the second most common polysaccharide in plants and is a family of low-molecular-weight polysaccharides integrated into plant cell walls. Hemicellulose structures work as matrix substances and framework forming separated wall layers. There is an important diversity of linkages and branching classes depending on the plant. Typically, this biopolymer is constructed from various sugar residues, such as D-mannose, D-xylose, or D-galactose, and has a shorter chain of sugar units (500 to 3000) than cellulose [33]. Normally, hemicellulose has less thermal and chemical stability than cellulose due to its lower polymerization degree and lack of crystallinity.Lignin is different from other natural biopolymers because its chemical structure differs due to the lack of glucose molecules that are replaced by aromatic heteropolymers. Lignin is a natural and amorphous biopolymer constituted by heterogeneous phenyl propane units. In the cell wall, it is linked together with different bio-compounds, cellulose, and hemicellulose, and acts as a barrier to enzymes, preventing them from entering lignocellulosic. The structure of lignin is responsible for its great resistance to chemical and enzymatic attacks. In addition, lignin is very relevant in the wood processing field and can be used to replace the usual organic material resources that are being exhausted, thanks to their properties, structure, and almost infinite reserve in the plants [34]. Lignin results from the oxidation of three phydroxycinnamyl precursors: Sinapyl, coniferyl, and p-coumaryl alcohols, also called monolignols. Lignins of different plant types show substantial differences in the proportion of these monolignols. Lignin extraction is quite difficult because the polymer is insoluble in most solvents, and its purification from plants requires complex chemical treatments with harsh settings, which can cause chemical modification or depolymerization. Therefore, commercial applications for lignin-based materials where complex chemical processes are not required are strongly desired [35].

### 2.4. Lignocellulosic Materials as Fillers

In lignocellulosic compounds, cell walls can be regarded as composites, because they are formed by fibers (cellulose), matrices (lignin, hemicellulose), and a variety of fillers (water and many organic compounds). The properties of lignocellulosic materials vary enormously between different plant species because they present important heterogeneity in their internal structures: Plant anatomy, cells, tissues, and organs. Since fibers or bundles are part of natural lignocellulose, the final mechanical behavior is governed by the chemical composition, internal architecture, and order of the bio-compounds [36].

Due to the increasing interest in eco-friendly and bio-based composites with better performance, the use of lignocellulosic fillers is being increased. Natural fibers and powders, in addition to their biodegradability and renewability, have the advantage of being cheaper, lighter, and more resistant per unit mass than the typical inorganic fillers (carbon black, zinc oxide, talc, or calcium carbonate) [37].

It is well known that the characteristics of nanocomposites are affected by the properties of their elements and by their interfacial and morphological characteristics. The use of reinforcing phase materials arises from natural and renewable sources in polymer composites have important advantages. The main qualities of lignocellulosic fillers in polymeric composites are: (a) Low cost, (b) low density, (c) renewable origin, (d) wide variability of structures, (e) chemically reactive surface, (f) relative high elastic modulus and strength, (g) relative easiness to degrade lignocellulosic composites by combustion (in comparison with inorganic fillers), (h) high sound attenuation capability, and (i) decrease in the machinery wear (because biobased resources are non-abrasive) [38,39].

There are examples of lignocellulosic fillers of multiple origins, such as sugar cane bagasse, jute, sisal, or wood. These materials from biomass can be introduced into various polymers of different natures. There are examples in thermoplastic polymers, such as poly(vinyl chloride (PVC), poly(lactic acid) (PLA), poly(ethylene) (PE), or poly(propylene) (PP); thermosetting resins, such as polyesters or epoxy resins, and rubbers (elastomers), as shown in Figure 4.

## 3. Thermoplastic Polymers

The main goal of this section is to study the state of the art of using olive stone powder residues as lignocellulosic fillers in thermoplastic polymers and rubbers. Polymeric matrices of polystyrene, polylactide, polyvinyl chloride, polypropylene, and poly(ε-caprolactone) were used to prepare composites filled with olive powder. These materials will be analyzed in terms of their mechanical, thermal, rheological properties as well as other complementary characteristics [40].

### 3.1. Polystyrene

Hamida et al. [41] studied different formulations containing polystyrene (PS)/olive stone flour (OSF) with various proportions by weight (0, 10, 20, and 30 wt.%). These composites were prepared by extrusion (50 rpm/min at 160–190 °C), followed by compression molding (170 °C, pressure of 300 bar for 10 min). Compared to the neat resin, authors obtained values around 14.4%, 10.62%, and 4.82% lower, respectively, for composites with 10, 20, and 30% wt.% of olive stone flour (OSF). This can be explained, in part, by the hydrophilic nature of the fillers that absorbed more moisture and caused a swelling in the polystyrene matrix (Figure 5a). This instigated the embrittlement of the material due to the volume occupied by the filler particles, creating defects. On the other hand, the OSF content influenced the evolution of the hardness, which increased with the addition of the filler. This result, and respective trend, would already be expected, because the hardness of the olive stone is superior to that of the resin.

### 3.2. Recycled Post-Consumer Plastic Material

La Mantia et al. [42] studied the effect of the addition of several organic fillers on the mechanical and rheological properties of a recycled post-consumer plastic material. For this purpose, a recycled polymer was used, which contained about 65–75% low density polyethylene (LDPE), 10–15% linear low-density polyethylene (LLDPE), 10–12% poly (ethylene-co-vinylacetate) (EVA), and small percentages of inert fillers and ultraviolet (UV) stabilizers. In terms of fillers, sago starch (SS), olive stones (OS) and sawdust (SD) were used. For comparison, a cheap inorganic filler, calcium carbonate, coated with stearic acid intended for better adhesion, was also used. Based on these materials, composites were produced with two compositions (30% and 60% by weight of filler), using a co-rotating intermeshing twin-screw extruder, and the specimens were obtained by injection molding. It was observed, regardless of the filler, that the flow curves of all composites with 60 wt.% were above the unfilled polymer, and relatively close to each other. In fact, the presence of inorganic fillers, in general, worsens the processability of the matrix due to increased viscosity. According to the authors, the viscosity of the composite filled by olive stone is higher than that observed for sago starch and sawdust, with a size between 0.25 and 1 mm, but lower than calcium carbonate (CaCO_3_) and sawdust, with size between 0 to 0.25 mm. In terms of tensile strength, except for the composite filled with CaCO_3_, all the others showed a decrease with increasing filler content. It was also possible to observe that the composite reinforced with olive stone, independently of the filler content, was the one that presented the lowest tensile strength. Similar behavior was observed for elongation at break, where for all fillers this property decreased with increasing the filler content. Therefore, some degree of stiffness was exhibited in all filled composites, leading to a more fragile material. In fact, authors obtained an increase in stiffness with higher filler contents, which explains the lower stress at break caused by the premature rupture of specimens containing organic fillers. Regarding the impact strength, authors found that fillers worsen this property, regardless of its concentration, due to poor adhesion with the nonpolar matrix. While the inorganic fillers show the best results, the olive stones present the best performance among the organic loads. Finally, a significant increase in the heat deflection temperature (HDT) was observed with the increasing of the filler content for all composites, and the values obtained are in good agreement with those reported for the elastic modulus. It was noted that the lowest value was obtained with fillers of olive stones.

### 3.3. Polylactide or Polylactic Acid

Koutsomitopoulou et al. [43] studied a novel biodegradable composite material, which involves olive pit powder dispersed into polylactic acid (PLA). Authors produced different types of PLA/pit composites, for which different particle sizes and filler content were considered. One group involved fillers with an average diameter of 30 μm and contents of 5 wt.% and 10 wt.%, while for the other two groups, the average diameters were 60 and 170 μm and had contents of 5 wt.%, 10 wt.%, 15 wt.%, and 20 wt.%. These composites were characterized in terms of physical and mechanical properties. From this study, the authors concluded that the size distribution and the physical/morphological characteristics of the granules depend on the grinding method used. Therefore, while the powders obtained by a high-speed rotor centrifugal mill showed similar fine distributions and a spherical shape, those obtained by a ball mill presented a wide distribution of size and granule-like flakes. Regarding the differential scanning calorimetry (DSC) analysis, authors noted an increase in the crystallinity ratio with the presence of the fillers, because they act as nucleating agents at lower concentrations and as obstacles for PLA chains to develop crystals at higher concentrations. In addition, a slight decrease in all transition temperatures was observed when the fillers were added. In terms of mechanical properties, and compared to the neat PLA, authors observed an increase of the tensile modulus with the filler content. For example, compared to the neat PLA and considering composites filled with 20 wt.% of olive pit powder, the improvements were around 18% and 16%, respectively, for particles with an average diameter of 60 and 170 μm. On the other hand, the tensile strength decreased slightly with the fillers content, which was explained by the physical and morphological characteristics of the granules resulting from the grinding method used and due to the poor interfacial bonding between olive pit powder and PLA. Similar behavior was noted for the bending properties. In fact, PLA is a brittle material, and when the fillers were added, they promote higher brittleness.

Perinović et al. [44] studied the thermal properties of poly(L-lactide)/olive stone flour (PLLA/OSF) composites. Differential scanning calorimetry (DSC) and thermogravimetric analysis (TGA) using dynamic heating regime were performed. PLLA was blended with 0, 10, 20, and 30 wt.% of OSF. Authors reported that the addition of olive stone flour (OSF) promoted the cold crystallization of PLLA. Therefore, the fillers (OSF) influence the crystallographic form and the morphology of the PLLA crystals, as well as the crystallinity degree. In addition, the processing procedure also affected the PLLA crystallinity. These authors assumed that the reinforcement acted as a nucleation agent at lower concentrations, but for higher concentrations, the olive stone flour behaved as an obstacle for polymer chains to make crystals. Regarding the PLLA glass transition, it was noted that the introduction of OSF into PLLA has a negligible influence on this parameter, which is indicative of the weak interactions between the polymeric matrix and the filler. From the scanning electron microscopy (SEM) analysis, they verified what would be expected, that is, the voids and agglomerates are more evident for higher levels of OSF content. Finally, the results of the thermal gravimetric analysis (TGA) for various heating rates demonstrated that the addition of OSF weakens the PLLA thermal stability, modifying the beginning of thermal degradation to lower temperatures. Moreover, the addition of OSF expands the PLLA degradation region and reduces the degradation rate. However, authors reported that the amount of OSF higher than 20 parts by weight had no significant influence on the degradation properties of PLLA (Figure 5b). The same authors, in another study [45], studied the crystallization behavior of semi-crystalline poly(L-lactide) (PLLA) incorporating olive stone flour (OSF) using differential scanning calorimetry (DSC). For this purpose, different composites were produced with various fillers content (100/0, 100/10, 100/20, and 100/30 wt.%), and the analysis performed during cooling of the melt at different and constant rates. Avrami, Ozawa, and Mo methods were used to obtain the kinetic parameters of crystallization. It was possible to conclude that the non-isothermal crystallization conditions applied to the melted samples lead to a complex crystallization process. In addition, it was found that these cooling conditions were not the only factor that affected the final crystalline properties of PLLA/OSF composites. In fact, the OSF fillers have a dual influence on PLLA, and they can act as a nucleating agent or as an obstacle to crystallization processes. On the other hand, to clarify the crystallization process of PLLA in the presence of OSF, non-isothermal crystallization kinetics was studied and, based on the Avrami equation, the data analysis indicated two crystallization mechanisms for all conditions analyzed: A complex nucleation and growth of PLLA crystals without and under the influence of OSF. Therefore, great attention must be paid to the crystallization process, when it is intended to produce composites with the desired properties. While the increase in cooling rates promotes a decrease in the size of the PLLA crystallite, a lower cooling rate is more suitable for obtaining regular lamellar structures.

### 3.4. Polyvinyl Chloride

Naghmouchi et al. [46] studied the addition of olive stone flour (OSF) up to 50 wt.% (20, 30, 40, and 50 wt.%) loading into a polyvinyl chloride (PVC) matrix (Figure 5c), in terms of mechanical performance, impact strength, water uptake, and wear resistance behavior. From the scanning electron microscopy (SEM) analysis, particles with sizes ranging from 20 to 150 µm were observed, which can be grouped into two populations. One exhibits a granular shape with a particle size of less than 50 µm, while the other group includes acicular particles with an aspect ratio of 4–10. In terms of mechanical properties, both tensile and flexural strength decrease with the filler content, but the biggest drop occurred for 20 wt.% of OSF followed only by a slight decrease until reaching 50 wt.% of olive stone flour, which is consequence of a low interfacial adhesion between filler and PVC matrix. For example, comparing the tensile and flexural strength between the neat polymer and the composite with the maximum filler content (50 wt.%), these authors observed, respectively, a decrease of 30% and 38%. The previously reported drop can be explained by the low aspect ratio of the OSF particles that act as stress concentrators instead of reinforcement filler. On the other hand, both tensile and flexural modulus increased with the increase of the filler content, which is due to the higher stiffness of the OSF filler rather than that of the PVC matrix. Using a Charpy impact machine, authors studied the impact strength, and this property decreased with the addition of the filler in both unnotched and notched samples, but with less effect for the last ones. This reduction was expected because the stiff fillers act as stress concentrators in the polymer matrix. Regarding the abrasion behavior, authors carried out Taber abraser tests and measured the weight loss after 1000 cycles. It was noted that weight loss increased almost linearly up to 40 wt.% of OSF, reaching values 50% higher than those obtained for neat resin, but for higher filler contents the weight loss increased dramatically, reaching values around 95% higher when 50 wt.% of OSF was added. This behavior is explained by the higher stiffness values obtained for higher values of filler contents. The water absorption properties were also evaluated and, from the curves obtained, authors observed that the water content increased initially very quickly, followed by gradual deceleration until reaching a plateau (constant value) after around 1 month of immersion into water. This is a consequence of the hydrophilic character of the filler associated with the existence of voids, pores, and cracks in the filler/matrix interface that promote the diffusion of water by capillary effect. Therefore, in this context, higher filler contents are responsible for higher values of water uptake. The mechanical properties were evaluated after immersion into water for 60 days, and the authors found a decrease in the tensile modulus and strength around 20% and 10%, respectively, in relation to the control samples. Finally, the thermal stability was studied, and authors found that the thermal degradation of OSF had three stages. Therefore, during the first stage, up to 200 °C, a weight reduction around 7% was observed due to the loss of moisture, but the real thermal degradation only started at 230 °C and continued until 300 °C, where there was a weight loss of around 60% due to the degradation of hemicelluloses. Finally, the third step occurred between 300 and 450 °C, with about 40% weight loss due to the thermal decomposition of cellulose and lignin. Regarding the PVC polymer, it was thermally stable until 240 °C, but, after this value, the real degradation was observed with two representative temperatures in the derivative thermogravimetric (DTG) curve. In this case, between 240 °C and 290 °C, there was a mass loss around 60% due to the dehydrochlorination of the PVC and volatilization of the hydrogen chloride molecules (HCl), while between 290 °C and 480 °C the mass loss was a consequence of the thermal cracking of the carbonaceous conjugated polyene sequences generated by the dehydrochlorination. On the other hand, the DTG curves evidenced three important temperatures (240 °C, 290 °C, and 450 °C), corresponding to the maximum weight loss rates, and three stages referring to the degradation process. In the first stage, the olive stone flour content added to the polymer decreased the degradation temperature and accelerated the initial degradation of the polyvinyl chloride matrix in the composite. The second stage, between 260 °C and 360 °C, occurred due to the dehydrochlorination of PVC and the degradation of hemicelluloses in olive stone flour. Finally, the third stage ranged from 360 °C to 480 °C, and was attributed to the carbonization process of the polymer. All the thermograms show a well-defined T_g_ around 75 °C with a position being insensitive to the filler content.

### 3.5. Polypropylene

Naghmouchi et al. [47,48] studied the use of olive stone flour (OSF) as a filler for a polypropylene (PP) matrix. Its characterization revealed that it consisted of acicular particles with sizes ranging from about 50 to 400 µm and composed of lignin, hemicelluloses, cellulose, and extract in contents of 35%, 35%, 25%, and 5%, respectively. For a filler content of 40 wt.%, these authors studied the effect of malleated polypropylene (MAPP) coupling agent amount on the tensile strength, tensile modulus, flexural strength, and flexural modulus. An increase in all properties was noted for higher amounts of MAPP up to 5%, a value from which no further improvement was observed. For example, the tensile and flexural strength were, respectively, around 75% and 210% higher than the values obtained for composites produced without MAPP. Regarding the tensile and flexural modulus, an increase of about 5–10% was noticed with the addition of 5% MAPP. These benefits were a consequence of the MAPP that improved the interfacial bonding between the filler and PP matrix, because the maleic anhydride (MA) groups chemically interacted with the filler by the formation of covalent bonds between the hydroxyl groups on the OSF surface and the anhydride groups of the coupling agent. These authors also studied the effect of OSF content on the mechanical properties of PP-based composites, produced by melt processing and injection molding, with filler contents up to 70 wt.%. In this context, they observed a linear increase of the tensile and flexural modulus with higher contents of olive stone flour (OSF), but, while this tendency was noted up to 70 wt.% for composites involving MAPP coupling agent, the maximum values were reached for 60 wt.% in composites without the coupling agent. In addition, they noticed a slight increase in modulus when MAPP was added. These benefits are due to the better dispersion of the OSF within the PP matrix when the coupling agent is added. In terms of tensile and flexural strength, both properties decreased significantly with the OSF content for composites without MAPP, while different behaviors were observed when the coupling agent was added. In this case, the tensile strength remained almost the same until 60 wt.% of OSF, followed by a decrease of about 10% for values of 70 wt.%. On the other hand, a continuous improvement was observed in terms of flexural strength until 60 wt.% of OSF, reaching values 36% higher than those observed for the neat polymer, followed by an abrupt decrease. This is a clear evidence that the composite is extremely sensitive to interfacial properties. Finally, it was noted that composites produced with MAPP showed a significant reduction in water absorption, which shows their sensitivity to the addition of a coupling agent.

Tasdemir et al. [49,50] produced two different polypropylene (PP) composites, where one of them was filled with olive pit and the other with almond shell flour (both between 0 and 40 wt.%), and the effect of load content on mechanical and morphological properties was studied for each material. In relation to composites filled with olive pits, the tensile modulus increased with the load content, reaching its maximum value for 40 wt.% of olive pit flour, which was 18% higher than the value obtained for pure PP. However, this value was 36% lower than that obtained for almond shell flour. On the other hand, the reverse trend was observed in terms of tensile strength and elongation at break, where both properties decreased with a higher filler content. For example, in relation to neat PP, there was a decrease of about 13.8% and 90.5%, respectively, in terms of tensile strength and elongation at break for composites with 40 wt.% of olive pit flour. This is explained by the agglomeration and poor dispersion of the fillers into the PP matrix. Finally, regarding hardness and impact strength, authors again observed an inverse trend. While the hardness increased with the filler content, reaching values around 8% higher for contents of 40 wt.% of olive pit, the Izod impact strength decreased about 82.5% for the same composite and compared to the values obtained for neat PP. This is a consequence of the weak interfacial adhesion between the olive pit and polymer that was confirmed by the scanning electron microscopy (SEM) analysis performed by the authors.

Gümüş et al. [51] produced polypropylene (PP) composites filled with olive pit particles in four different concentrations (5, 10, 15, and 20 wt.%) and maleic anhydride as modifier. Authors reported that PP’s spectra contained some group frequencies (involving CH_2_ and CH_3_ groups) and the peaks in the spectrum are visible at 2916 cm^−1^ (CH_2_), 2959 cm^−1^ (CH_3_), 2841 cm^−1^ (CH_2_), 1460 cm^−1^ (CH_2_), 1376 cm^−1^ (CH_3_), 1357 cm^−1^ (CH_2_–CH), 1328 cm^−1^ (CH_2_–CH), 1302 cm^−1^, 1224 cm^−1^, 1260 cm^−1^, 1170 cm^−1^, and 1153 cm^−1^ (CH_3_), (CH_2_), and (CH), respectively. Regarding the PP Fourier transform infrared spectroscopy (FTIR) spectra, it was possible to report one peak at about 1550 cm^−1^ bands due to the lignocellulosic bonds after an added 15% ratio of olive pits, and a linear correlation was also observed between the olive pits amount and the peak intensity of lignocellulosic material. In this context, the addition of the olive pit particles did not distort the spectrum, however, a decrease of 5% occurred in the intensities in the 2916 cm^−1^ (CH_2_), 2959 cm^−1^ (CH_3_), 2841 cm^−1^ (CH_2_), and 1376 cm^−1^ (CH_3_) regions. The carbon–hydrogen bonds were physically affected by the amount of olive pit particles, therefore, and due to the differently sized particles, the peak intensity changed. Authors also analyzed the manufactured composites’ surfaces morphology, and it was noticed that the PP with 0 to 15 wt.% presented a homogeneous and smooth surface structure. Cracks and air bubbles were not found in all specimens with filler content on that range. However, a rougher and more irregular surface was reported in the PP filled by 20 wt.% of olive pit particles composite. In addition, authors used the SEM analysis to conclude that the addition of olive pit particles to the PP matrix did not cause a deformation on the surface and that, in addition to the filler being well dispersed into the matrix, there was a good adhesion between filler and matrix. Finally, to study the evaluation of the oxidative stability, authors preformed Oxygen Induction Time (OIT) tests on the manufactured composites. It was noticed that the OIT’s value of neat PP was 1.15 min, however, when 20 wt.% of olive pit particles were added to the PP, the OIT value reached 30.1 min. Through OIT’s data, the PP without the reinforcement was not able to effectively protect the material against thermo-oxidative conditions, nevertheless, when the olive pit was added to the polymer, the OIT values increased significantly. The results also indicated that the material reinforced with olive pit powder was more resistant to high temperatures than neat PP. From the thermal gravimetric (TGA) tests, the thermal characteristics of the composites showed that increasing the olive pit powder content, for a given temperature, the decomposition rate decreased. In this context, increasing the temperature from 200 °C to 800 °C, the weight losses were 99.4% for neat PP, 96.7% for 5.0 wt.% PP/olive pit composite, 94.9% for 10.0 wt.% PP/olive pit composite, 89.5% for 15.0 wt.% PP/olive pit composite, and 87.2% for 20.0 wt.% PP/olive pit composite. Therefore, with the addition of the olive pit powder, the decomposition behavior did not change, but the curves shifted to higher temperatures, showing, in this case, that the olive stone particles improved the thermal stability of PP in relation to untreated PP. Lastly, authors observed that the olive pit particle–treated PP composite provided higher storage modulus than the untreated one due to the better interfacial adhesion between the olive pit particles and the polymeric matrix. For all tested specimens, the storage modulus decreased with the increase of temperature. The results of the dynamic mechanical analysis (DMA) demonstrated that the storage modulus of the composite increased with the addition of olive pit particle. Regarding the loss modulus of the manufactured composites, the effect of the filler was similar to the storage modulus—it increased with the addition of olive pits. With these results, authors could conclude that, with the increase in olive pit content, the damping properties of composites improved.

### 3.6. Poly(ε-Caprolactone)

Hejna et at. [52] used olive stones (OS), date seed (DS), and wheat bran (WB) as fillers for biocomposites based on poly(ε-caprolactone) (PCL) and, for this purpose, they produced different samples denominated as PCL/OSX, PCL/DSX, or PCL/WBX, where X stood for filler used in parts by weight (0, 10, 30 and 50 pbw). Figure 6 and Figure 7 show, respectively, the particles size and the macroscopic appearance of the fillers.

From the SEM analysis, authors observed that the largest filler particles were observed for PCL/WB composites. Therefore, due to its size, high porosity, and absence of strong interfacial interactions, this group of materials showed the least compatibility among all. Consequently, a noticeable drop in tensile strength and elongation at break was observed. On the other hand, compared to OS and DS fillers, the addition of WB resulted in a higher decrease in tensile strength, which can be explained by the lower content of polysaccharides (cellulose, hemicellulose, lignin, or starch) that enhances the interfacial interactions with the matrix due to the large number of polar groups. In addition, WB has a higher protein content, which can act as plasticizers in the PCL matrix. Furthermore, this explains the highest elongation at break of PCL/WB composites.

Regarding the composites with higher filling contents, they showed significant dependence on the type of filling. Therefore, the reduction in mechanical performance was instead associated to the disturbance of matrix integrity and limitation of PCL chains mobility. The stiffening of the material, expressed by increasing the modulus, was due to the incorporation of lignocellulosic fillers. This was related to the mobility of the PCL chains limited by filler particles and potential interactions between the PCL carbonyl group and polar groups present on the surface of applied fillers.

Authors confirmed that the addition of these fillers into poly(ε-caprolactone) matrix resulted in increased hardness. The PCL/OS composites showed the most significant increase in hardness, from 50.2 for neat PCL to 52.8, 54.7, and 56.7 °Sh D, respectively, for 10, 30, and 50 pbw of filler. For materials containing WB, the lowest values of hardness were found (around 5% lower than for PCL/OS composites). Finally, these authors observed some differences in T_g_ between composites with different fillers. In this context, materials containing lignocellulosic fillers exhibited significantly higher T_g_ values (difference of almost 6 °C for PCL/OS50 composite) compared to neat poly(ε-caprolactone). The lowest T_g_ values were associated to PCL/WB materials due to the existence of amino acids, which increase the PCL’s ability to flow under stress. Regarding the water uptake values, it was observed that the composites containing WB were characterized by a water absorption more than 50% higher than the PCL/OS and PCL/DS materials.

## 4. Thermosetting Resins

The application of lignocellulosic fillers to thermosetting resins was considered an attractive alternative to increase the properties of such materials or to add new capabilities [53]. Phenol-formaldehyde, unsaturated polyesters, and epoxy resins are the most common options for high-performance applications [54], and the respective chemical structures are shown in Figure 8.

### 4.1. Phenol-Formaldehyde Resins

Simitzis et al. [55] prepared and studied carbonaceous adsorbent materials from novolac resin, a phenol-formaldehyde resin type, and pressed olive stones. They analyzed the structure, specific surface area, adsorption characteristics, and adsorption capacity of mixtures of phenol-formaldehyde-cured resin with pressed olive stones in a weight ratio of 20:80. The thermosetting material (novolac) was obtained after the polymerization reaction of phenol with formaldehyde in proportions 1.22:1 (mol:mol) with oxalic acid as catalyst in proportion 1.5% w:w phenol. After that, the resin was separated, dried, and pulverized. Pressed olive stones were grounded and sifted until homogeneous grains of less than 300 µm were obtained. The two-component mixture was combined in 20:80 weight ratio novolac:pressed olive stones and cured at 170 °C for 30 min in cylindrical molds. Finally, the resulting composites were carbonized in N_2_ atmosphere. The adsorption ability of the prepared samples was compared with two commercial activated carbons. As chemical materials for testing, two phenols were chosen: M-nitrophenol and p-nitrophenol, because they are common in water purification and they can work as model-substance in the porous structure of the adsorptive materials. For comparison, all materials were grounded and sifted to obtain diameters less than 63 µm, and then the samples were mixed with an aqueous solution (0.032 g/L) that contained one of the organic pollutants and, after different intervals of time, they were analyzed.

It was observed that the pores of the laboratory samples were different from the commercial ones. The olive derived resemble charcoals treated with steam, retaining the wood morphology. In addition, the influence of the sample and grains shape and size in the adsorption was found. The adsorption capacity was higher for the samples having the shape of small cylinder grains in relation to the corresponding powdered samples. The adsorption of the different organic compounds on activated carbons followed the Freundlich-equation or the Langmuir-equations. Results showed very good adsorption of phenol, 3-nitrophenol, or 4-nitrophenol from aqueous solution in the carbonaceous materials by activation and in the commercial ones. The adsorption capability of the three phenols examined in the pulverized materials followed this order: P-nitrophenol > m-nitrophenol > phenol.

In another study published by the same authors [56], they continued to prepare mixtures of novolac resin and olive stone biomass in proportions 20:80 (wt.%:wt.%) which were first cured and then pyrolyzed up to 1000 °C (material C20) and, finally, activated with steam (material C20a). The sample manufacturing process was very similar, but, in the present case, the carbonization of the cured materials was carried out in cylindrical tube ovens under continuous flow of N_2_. First, the furnace was heated to 1000 °C at a rate of 4 °C/min, followed by an isotherm for 10 min, cooled to room temperature and, finally, kept under vacuum. In addition, the pyrolysis residues were activated with steam at 930 °C. The adsorption properties of the materials (C20 and C20a) and a commercial activated carbon (CC) were investigated based on the adsorption of pentane and nitrogen capability. The adsorption capacity and the surface area were determined following the Brunauer–Emmett–Teller (BET) and Dubinin–Radushkevich–Kaganer (DRK) equations. The pore volume was measured as micropore volume employing the Dubinin–Radushkevich (DR) equation and using the Kelvin equation as cumulative pore volume. In this context, the results obtained that the adsorption properties measured based on the adsorption of nitrogen and pentane, i.e., pore volume, surface area, and adsorption capacity, followed the next sequence: C20a > CC > C20. The activated materials C20a and CC were principally microporous and in the Brunauer classification for nitrogen adsorption were part of the type I isotherms. However, the material C20 contained part of microporosity and mesoporosity, which explains being part of types I and II. The DR equation could be applied for the adsorption of nitrogen in all materials in the region of the relative pressure P/P0 = 0.005–0.3 [36].

Similar studies involving the same material (novolac resin and olive stone biomass in proportions 20:80 (wt.%:wt.%) were developed by Simitzis et al. [57] but, in this case, the thermal treatments were different for each type of sample. Material C2 was cured and pyrolyzed up to 800 °C, while material C3 up to 1000 °C and activated with steam (C4). Further, material C1 (100% olive stone biomass) was also pyrolyzed up to 1000 °C. Three characterization methods were used: (a) Mercury porosimetry; (b) adsorption of different organic adsorbates from the vapor phase: Pentane, cyclohexane, and toluene; and (c) adsorption of methylene blue dye from aqueous solution. From these tests, the results obtained showed that although the weight losses due to reactions during carbonization were very high in the range of 200–600 °C, they were very low above 800 °C, and the material pyrolyzed at 1000 °C was more stable in the pyrolysis. The different reactions affected the pores formation, as it could be observed in the porosimetry results. A continuous drop in the pore volume is detected in the range of 10 000–72 570 Å from C2 to C3 and to C4. For the C3 the pore volume that was in the range of 37–100, Å decreases slightly (and increases barely for 10–10,000 Å), and for C4, the pore volume rises in the range of 37–100 Å. In the adsorption of different organic adsorbates experiments, different molecules with different molecular diameters, polarities, and conformations were used. Sample C1 showed low adsorption capacity and pore volume, because it contained few micropores, whereas the pore volume and adsorption capacity increased for the other samples, in the order C1 < C2 < C3 < C4. Micropores were also contained at 800 °C (C2), but more were formed during the pyrolysis at 1000 °C (C3), and their formation was specially favored by the action process (C4). In the case of adsorption of methylene blue, in general and dyes have larger molecules than the vapors, between 1.30 and 1.35 nm^2^. Adsorption of dyes requires pores 65 times larger than the particles, i.e., 20 to 500 Å. The results determined that C1 had a high content of mesopores that favored the methylene blue adsorption. The others, mesopores and micropores, facilitated the adsorption of the dye and vapors, respectively [37]. In summary, it was found that Cl contained a majority of mesopores and few micropores, because it adsorbed methylene blue well, but not the vapor elements. C2, C3, and C4 had micropores (C2 < C3 < C4), but also mesopores, and adsorbed both vapor (organic compounds) and liquid phase (dye). These results were better than the commercial alternatives and the versatility of the developed manufacturing process allowed to obtain activated carbon materials with a desired pore structure and adsorption characteristics.

Sfyrakis et al. [58] studied the effect of different pyrolysis temperatures in the novolac:/olive stones 20:80 (wt.%:wt.%) compositions. The carbonization of the cured specimens was carried out in a cylindrical furnace heated to different temperatures, between 500 and 1000 °C at a rate of 4 °C/min, followed by an isotherm of 10 min, and, finally, cooling to room temperature under vacuum. The weight losses and the shrinkages taking place in the carbonization process increase up to 600 °C. The pyrolysis residues were analyzed, and the ability to adsorb toluene and cyclohexane from the vapor phase was qualitatively similar, because both were slightly adsorbed on the adsorbents pyrolyzed up to 600 °C. The adsorption of toluene is remarkably higher than that of cyclohexane for the corresponding adsorbents pyrolyzed above 800 °C. Above 800 °C, the adsorption of toluene was much higher than that of cyclohexane. For the cyclohexane and toluene, the relative order of the absorbents with respect to increasing vapor uptake was the same: (1) Very low at 500–600 °C; (2) intermediate at 800 °C; and (3) higher between 900–1000 °C. In addition, higher values were observed for toluene than cyclohexane for the adsorption rate, equilibrium, and uptake values, meaning that the samples (adsorbents) have good selectivity for the toluene. There are many parameters that affect the adsorption capability, such as the polarity, size, shape, and distribution of the pores; but normally it is quite difficult to evaluate them. Between 500 and 600 °C, high weight loss was attributed to the material shrinkage (reduction of dimension) that led to the formation of macropores and eased the diffusion of toluene. For the cyclohexane, some deviations were detected at 600–800 °C that were attributed to its different polarity, molecular conformations, and dispersion forces between adsorbent and adsorbate. Furthermore, the adsorbent pyrolyzed at higher temperatures (1000 °C) presented slit-shaped pores, which are typical of carbon molecular sieves [38].

Gil et al. [59] produced composites with phenol–formaldehyde resins filled by olive stones (OS). Those materials were employed as precursor materials for the preparation of microporous activated carbons for use in the separation of CO_2_. In this study, two phenol–formaldehyde resins were synthesized (Resol and Novolac). In the first resin, Resol, a 2.5:1 formaldehyde-to-phenol ratio was used to obtain the Re resin sample. The second type of resin, Novolac, two formaldehyde-to-phenol mole ratios were used, 1:1 and 1:1.22, to yield the No1 and No2 resin samples, respectively. Incorporating potassium chloride (KCl) to the Re and No1 cured resins, the carbon precursors were prepared. The cured resins were impregnated with KCl at room temperature (ReKCla and No1KCla precursors) or boiled with a saturated KCl solution (ReKClb and No1KClb precursors). Nevertheless, the No2 cured resin was used as additive to the olive stones in a 80:20 wt., with a proportion of OS:resin. This blend was then mixed with hexamethylenetetramine (28.6%) and heated at 170 °C for 30 min (No2OS precursor). The precursors were then carbonized in a horizontal furnace under a nitrogen flow rate of 50 mL.min^−1^. Prepared at room temperature, the ReKCla and No1KCla precursors were carbonized at 600 °C and produced ReKCla-600 and No1KCla-600 carbonized samples. Prepared at a boiling temperature, the ReKClb and No1KClb precursors were carbonized at 1000 °C and produced ReKClb-1000 and No1KClb-1000 carbonized samples. Lastly, the No2OS precursor was carbonized at 1000 °C to obtain the No2OS-1000 carbonized sample. It was reported that the interaction between temperature (T) and burn-off (B) was detected in the experimental region under consideration. At high activation temperatures (T > 700 °C for ReKCla-600 and T > 810 °C for No1KCla-600), authors reported that CO_2_ uptake decreased with the increase in burn-off. However, at low temperatures, the burn-off had not significantly influenced CO_2_ uptake. Moreover, over the temperature range studied and in the absence of interaction, the effect of burn-off was low or null. Authors used the response surface methodology (RSM) to evaluate the combined effect of activation temperature and burn-off degree on the CO_2_ capture capacity of phenol–formaldehyde resin and olive stone-based activated carbons. In the experimental region studied, the activation parameters do not similarly influence the capture capacity of all the evaluated materials. Therefore, these authors concluded that No1KCla-600 reached a maximum CO_2_ capture capacity of 9.3 wt.% at an activation temperature of 809 °C and a burn-off of 22%. The CO_2_ adsorption capacity values (at 35 °C and atmospheric pressure) of a mixture of this type of resin (20 wt.%) with olive stones (80 wt.%) were similar to those of commercial activated carbons (7.3–9.3 wt.%). Furthermore, for samples No1KClb-1000 and No2OS-1000, the greatest CO_2_ uptakes (7.5 and 7.3 wt.%, respectively) were achieved at 800 and 942 °C, respectively, regardless of the burn-off degree. They reported that great potential as adsorbents for CO_2_ capture at atmospheric pressure was showed by activated carbons derived from the Novolac phenol–formaldehyde resin type and from OS [39].

### 4.2. Unsaturated Polyester Resins

Elsahli et al. [60] produced particleboards, which were characterized in terms of mechanical and physical properties. Particleboards are used in non-structural applications, such as furniture, shelves, or doors. Usually, they are uniform pieces with different configurations, sizes, or thicknesses. In their production, phenol, urea, melamine formaldehyde, polyester, or epoxy resins with lignocellulosic bioproducts are used [40], but in this study, the authors used an unsaturated polyester resin filled with olive stones. Several olive stone/polyester compositions were tested, like 52:48, 60:40, and 80:20 wt.%:wt.%. After the olive stones were cleaned, washed, and dried under the sun for 48 h, the resin was added. Then, 8 tons of pressure for 10 min and a heating treatment of 70 °C for 15–20 min were applied.

These particleboards using olive stone waste were characterized according to European standards, and most of the results were in accordance with the requirements of American, British, and European standards. For example, considering the particleboard 80:20 wt.%:wt.%, the results showed a porosity around 14.21%, particle size of 22.06 cm^3^, particle density around 1.39%, size dimensions 3.65%, permeability 12.70 mD, and the total Fungi and bacteria account were Absent (cfu/g). These results demonstrated the potential of olive stones in the manufacture of agglomerated panels with adequate mechanical and physical properties.

Gharbi et al. [61] produced composites by compression molding (at 110 °C) using untreated olive nuts flour (ONF) and γ-mercaptopropyltrimethoxysilane (MRPS)—modified ONF. For this study, authors used a polyester resin matrix with 10 to 60 wt.% of ONF. In terms of flexural modulus, Figure 9 shows an increase with the increase of ONF up to 55 wt.% for both composites.

Compared to neat resin, composites with 10, 30, 40, and 55 wt.% of ONF exhibited flexural modulus around 1.36, 1.9, 2.07, and 2.51 times higher, respectively. However, the same composites with a silane treatment exhibited flexural modulus around 1.45, 2.2, 2.5, and 2.8 times higher, denoting a significant improvement with the silane-based treatment.

Regarding flexural strength, Figure 10 shows different behaviors. For composites with untreated fillers, the flexural strength increases slightly above 42 MPa, a value observed for neat resin, reaching a maximum strength around 48 MPa for 40 wt.% of ONF, from which it starts to decrease. However, this property increased with the treatment, reaching, in this case, a maximum strength around 51 MPa for 30 wt.% of ONF, from which it starts to decrease as previously verified. Authors concluded that the improvement observed when the ONF was treated resulted from the improvement of the interfacial adhesion between matrix and fillers. Finally, there was a significant reduction of the Charpy impact strength with the filler content. This trend was expected because, as the filler is stiffer, it acts as a stress concentrator in the polymer matrix and, in this context, facilitates the crack initiation. Moreover, authors reported that MRPS treatment was ineffective to modify this trend.

The improvement of interfacial adhesion after silane modification was evaluated by dynamic mechanical analysis (DMA) and SEM observations. From Figure 11 and for the untreated ONF, clean surfaces with cavity gaps are observed between the fillers and matrix, while for the ONF treated with the silane coupling agent, a better contact between filler/matrix is noticeable, promoting, in this case, better interfacial adhesion between the two phases.

Authors also studied the water absorption properties and observed that the water content increased rapidly in the first absorption phase, followed by a gradual decrease until reaching the saturation plateau after about 1 month of immersion into water. However, the water absorption increased for higher contents of ONF, but when the fillers were treated, the water absorption decreased. For example, the maximum water absorption decreased from around 9% to about 5%, considering composites with 40 wt.% of untreated and treated ONF, respectively. The increase in interfacial adhesion between the filler and the matrix, as well as the improvement in the wetting of the ONF particles by the polymer, are the causes of this reduction. Another reason that contributes to the decrease in the water sensitivity of the composite is the decrease in the hydrophilic character of the filler, after treatment with MRPS.

### 4.3. Epoxy Resins

Papanicolaou et al. [62] studied the effect of thermal shock cycling on the mechanical behavior of epoxy–matrix (RenLam CY219) composites reinforced with olive pit powder. Authors manufactured neat epoxy resin specimens and composites reinforced with olive pit for different volume fractions (V_f_ = 7, 14, 27 and 44% of olive pit’s powder). After being subjected to thermal shock cycling conditions in the temperature range from −27 °C to +80 °C, the specimens were properly characterized. Normally, when the filler content increases, the strength of particulate polymers decreases due to the creation of aggregates, however, for the manufactured materials, only a small decrease in the flexural strength has been reported for larger filler-volume fractions. On the other hand, the addition of olive pit powder to the neat resin led to an increase in the flexural modulus. In this case, they observed that the flexural modulus increased with the fillers content, reaching a maximum value of around 48%, compared to neat resin, with a V_f_ = 44% of olive pit powder. These benefits result from the good adhesion between filler and matrix, as reported in the SEM micrographs obtained by the authors. In addition, to predict the modulus and compare the predicted values with the experimental ones, authors used the modulus predictive model (MPM). It was found that the degree of adherence and degree of dispersion were k = 0.357 and λ = 0.355, respectively, in addition to excellent repeatability, because the experimental deviations were less than 3%. Therefore, the modulus variation with the reinforcement content can be effectively predicted with the modulus predictive model (MPM). On the other hand, to predict the modulus variation as a function of the number of thermal shock cycles, and different filler contents, the residual property model (RPM) was used. However, different damage mechanisms occurred within the materials, due to their heterogeneous nature and very dissimilar expansion/contraction behavior from the constituents, which resulted in the degradation of their physical and mechanical properties. Therefore, for application of RPM, P_∞_ was obtained for composites with V_f_ = 7, 14 and 27% of olive pit powder, from the experimental flexural modulus at 200 cycles, and for composites with V_f_ = 44% at 100 cycles. P_0_ was obtained from the experimental flexural modulus of virgin composites. The temperature range of thermal shock cycling, ΔT, was 107 °C. The experimental results and respective predictions showed a good agreement between them, in most cases. However, the flexural modulus of the materials produced remained almost unchanged, even under the hostile environment, for all the cases studied.

In another study, Papanicolaou et al. [63] developed an environmentally friendly epoxy adhesive using dry olive pit powder with 0 to 3.5 wt.% of filler content. A mill was used to transform the olive pits into fine powder, reaching a maximum particle diameter of 5 µm. Lap joint specimens were prepared using aluminum adherends. From this study, it was proven that is possible to produce a composite adhesive with matrix and reinforcement environmentally friendly, promoting a 20% increase in shear modulus compared to the neat matrix. However, further investigation is suggested by the authors to obtain improved systems.

Finally, in another study, Papanicolaou et al. [64] produced composites reinforced with olive kernel pyrolytic, and the influence of the particle-weight fraction (0–25 wt.%) on the flexural behavior of such composites was studied. For this purpose, polymeric composites were prepared by mixing pyrolytic char of olive kernels with an epoxy resin (RENLAM CY 219 with RENLAM HY 5161 as a hardener). Olive kernel is a lignocellulosic biomass, which is typically composed by cellulose, hemicellulose, and lignin. Authors reported that the carbon percentage in the pyrolytic char was around 66.98 wt.% and the hydrogen about 1 wt.%. They also performed a metal analysis and concluded that the raw material has a high Fe content. In terms of mechanical tests, the results showed that the maximum strength decreased up to 10% of reinforcement, followed by a slight increase. However, compared to non-reinforced material, this property has always remained less. This can be explained by two phenomena. The first one is due to the low adhesion between the reinforcement particles and the polymeric matrix, promoting, in this case, interfacial voids, while the second is that the addition of filler resulted in a partial reinforcement effect that was stronger with increasing filler concentration. On the other hand, it was reported that the flexural modulus increased continuously, however, after 0.313 V_f_, it remained almost constant.

Erkliğ et al. [65] studied the incorporation of olive pomace (OP) in composites reinforced with glass fiber and an epoxy resin (Figure 12). Different weight ratios (0.5, 1, 2, 5, 10, and 15 wt.%) of OP particles were incorporated into the epoxy resin to investigate their effects on the tensile, flexural and damping properties of glass fiber reinforced polymer (GFRP) composites. The grain size of the OP particles was 75 μm, after the sieving process.

The highest benefit obtained in terms of tensile strength was achieved for 5 wt.% of OP particles, evidencing an improvement of 16.6% compared to composites with neat resin (from 319.5 MPa to 372.65 MPa), while the flexural strength reached an improvement of 35.5% (from 465.56 MPa to 630.9 MPa). It was also found that the addition of particles promoted an increase in maximum elongation at the break point, however, the addition of OP particles above 5% (ideal value) impaired chemical compatibility and charge transfer between the matrix and the particles. In terms of damping ratio, the highest value was obtained for 10% by weight of OP particles, a value that was about 70% higher than that obtained for composites with neat resin. Finally, the maximum storage modulus was reached for 10 wt.% of OP filler, evidencing an improvement of 74.2% in relation to the value obtained for composites with neat resin. This is a consequence of the good chemical compatibility between the OP particles and the epoxy resin and, in addition, the incorporation of the particles contributed to the increase in the dissipated energy.

Mohammed et al. [66] studied the effect of date seeds (DS) and olive seeds (OS) on the wear rate and hardness of an epoxy resin. These composites were produced, using the manual layup technique, with 0, 8, 13, and 18 wt.% of fillers content and with a granulometry of 300, 450, and 600 µm. The results revealed that the hardness increased with the increase of the filler content, while the wear rate decreased. Furthermore, regardless of the filler used, the best hardness and wear resistance values were obtained for 18 wt.% and a particle size of 300 µm. In terms of hardness, and through a mathematical model, these authors also concluded that the filler weight content has a greater effect on this property than the particle size.

## 5. Rubbers or Elastomeric Polymers

Finally, rubbers have great commercial success due to their excellent chemical and mechanical behavior, such as resistance to crack propagation, elasticity, or flexibility. Rubbers can be of synthetic or natural origin, but both after the vulcanization process are being used in a multitude of products, such as gloves or tires [67].

Khalil et al. [68] investigated composites based on acrylonitrile butadiene rubber/devulcanized waste rubber (NBR/DWR) blends containing several concentrations of olive stones waste (OSW) obtained by conventional blending (Figure 13). The filler treatment was conducted by impregnation in acrylate monomer, trimethyl-propane trimethacrylate (TMPTMA) under gamma irradiation, and the results compared to those obtained with untreated fillers. Initially, authors produced samples with different proportions of NBR/DWR (100/0; 90/10; 80/20; 70/30; 60/40 wt.%:wt.%) and the architecture responsible by the highest mechanical properties (70/30 wt.%:wt.%) was subsequently analyzed with different concentrations of OSW (5, 10, 20, and 40 phr).

Authors observed that the tensile strength (TS) decreases with the addition of 10% DWR to the NBR, however, for higher DWR contents, this property increases systematically, reaching maximum values for 30% DWR. This increase was attributed to the nature of NBR as non-crystallizing, with low gum strength. With the addition of 40%, a further decrease in TS was observed. In terms of elongation at break (E_b_), this mechanical property increased with the addition of DWR to NBR up to 20% of DWR. However, in blend 70/30 NBR/DWR, E_b_ values decreased due to the good compatibility of DWR with NBR, as previously revealed for the tensile strength. Finally, the modulus increased to DWR levels between 10 and 30%, reaching the maximum to 30%, but values above 30% of DWR led to its decrease. Therefore, as 70/30 NBR/DWR showed the best results for tensile properties, authors used this combination for further investigations and added various OSW concentrations (between 5 to 40 phr). Figure 14 shows the tensile results obtained for 70/30 NBR/DWR blends filled with different treated and untreated OSW contents.

It is possible to observe that the tensile strength increases with the increase of olive stones waste (OSW) up to 10 phr, for both untreated and treated fillers, but, posteriorly, this value decreased until reaching the maximum concentration (40 phr). In addition, when compared with OSW, and for the same concentration, composites with treated olive stone waste (OSWT) showed better values for tensile strength (TS). The composite with 10 phr OSW/OSWT achieved the highest tensile strength due to the presence of hydroxyl (OH) groups, because olive stones waste has a hydrophilic nature. The interaction of these groups with the NBR nitrile group increased the TS values, however, there was a reduction in the TS to values above 10 phr due to the agglomeration of the fillers. Nevertheless, when increasing the OSW content, the elongation at break (E_b_) values decreased for both fillers, although the composites filled with OSWT showed the lowest values.

This can be attributed to the higher stiffness of OWS fillers than to the blend. In addition, when compared to OWS, the treated filler adhered well to the blend. Finally, in terms of modulus, it was found that it increases with the increase of OSW or OSWT contents up to 10 phr, a value from which it starts to decrease. When compared to OWS, at the same concentration, OWST composites present higher modulus due to the increase in interfacial adhesion, as a result of surface treatment. The ideal combination for tensile strength and modulus has been achieved by 10 phr.

In the same study, authors investigated the water uptake, for both composites (treated and untreated olive stone waste) at room temperature, and the results are shown in Figure 15. Due to the hydrogen bonds with water, resulting from the presence of polar NBR, it is possible to observe a higher water content in the composite for higher concentrations (wt.%) of fillers. However, the treated fillers showed less water absorption. Therefore, while the hydrogen bonds between the hydroxyl (OH) groups of cellulose in the olive core and the water molecules explain the increase in water content, grafting with acrylate monomer that covers the surface of the olive core leads to a decrease in free volume and voids, with the consequent reduction of the water content in the treated composites.

Finally, Figure 16 shows the abrasion effect of the different materials analyzed. It is noted that the abrasion loss was greater in the composites than in the NBR/DWR, which can be explained by the greater fragility of the OWS distributed in the NBR/DWR.

## 6. Synthetic Summary

A review of all published works was presented in the previous section, where the benefits obtained from the olive pits are reported in detail. Therefore, this section intends to summarize all the benefits reported and, for this purpose, Table 3 presents them according to the type of polymeric matrix used (thermoplastic, thermosetting, or elastomeric), main properties obtained/characterized, and main application for all material and/or objectives of the scientific work.

## 7. Discussion

Olive oil has been a fundamental product for Mediterranean countries and culture. Initially obtained by traditional and few ecological methods, they have been gradually transformed and, nowadays, they are industrialized processes of high technology [69]. However, this industrialization promotes a significant increase in biomass byproducts, with consequent management and processing difficulties. On the other hand, the legislation imposed to the olive oil sector [23], with the aim of making it more ecological, and the current green policies resulting from a more conscious society lead to the economic and environmental valorization of this waste.

The olive fruit presents three identifiable sections: Skin, pulp, and stone. After the olive oil manufacturing process, from all residues, the olive stone is the most abundant. In addition, the table olive industry also participates with a large amount of this waste. This residue is considered a lignocellulosic material, the most available in the biosphere, which is formed by three chemical compounds: Cellulose, hemicellulose, and lignin. Basically, cellulose is a chain of glucose molecules connected by β(1→4)-linkers. Hemicellulose is formed by various types of sugars of 5- and 6-carbons; and lignin is synthesized by polymerization of three phenolic molecules. All three chemical compounds are organized in supramolecular structures during millions of years of natural evolution, presenting a complex organization into microfibrils that integrate structural stability in the plant cell walls. However, the great structural and chemical stability of lignocellulosic biomass hinders possible treatments as waste. Nevertheless, using chemical and/or physical treatments to purify and separate the individual components of the biomass allow them to be transformed into different molecules for the chemical industry. Although viable, these processes are difficult, expensive, and present environmental problems [6,7].

Therefore, there is an interest in direct applications and without any complex treatment. In this context, taking advantage of the benefits of lignocellulose, the literature reports its integration into polymers (thermoplastics, thermosets, and rubbers) in form of fillers [27,28]. In this way, it is economically valued and, simultaneously, the environmental impacts of this residue and synthetic polymers are minimized at the same time. This scenario has been encouraged in the field of materials technology and is in line with Council Directive 91/156/EEC, which suggests the adoption of measures to limit the production of waste, namely through the promotion of clean technologies and products, taking into account the existing or potential market opportunities for recovered waste [70].

In this context, and in general, literature review shows an increase in the elastic modulus with the olive stone content, while the tensile strength and elongation at break decrease. Similar behavior is observed for the same properties obtained in the bending mode, but this loading mode is shown to be much more sensitive to the interfacial properties. Both tensile and bending strength decrease due to the insufficient adhesion between fillers/matrix and, when the volume fraction increases, more interfaces and cavities are formed in the composite [16,71,72]. In fact, surface treatments improve the interfacial adhesion between fillers and matrix due to the formation of covalent bonds between the hydroxyl groups on the surface of the filler and the anhydride groups of the coupling agent. Regarding the impact strength, fillers based on olive stones worsen this property, because the stiff filler act as stress concentrators in the polymer matrix. Mechanical properties also decrease in presence of humid environments due to the hydrophilic character of the olive stone associated with the existence of voids, pores, and cracks in the filler/matrix interface that promote the diffusion of water by capillary effect [46]. Therefore, higher filler contents are responsible for higher values of water uptake. In terms of abrasion behavior, the weight loss decreases with the filler content and the opposite trend occurs regarding the hardness. This behavior is explained by the higher stiffness values obtained for higher values of filler contents.

Therefore, from the present review, it is clear that polymeric composites based on olive stone fillers have a high potential in a wide spectrum of engineering applications. At the same time, it seems clear that additional efforts must be made to understand the mechanical behavior (fracture and fatigue behavior) of these materials with a view to their application in engineering.

## 8. Conclusions

From this review about olive stones as a filler for polymer-based composites, several conclusions can be drawn, especially for future works:-The various studies carried out with different types of polymer matrices demonstrate the interest of the scientific community in the reuse of olive stone residues in polymeric composites;-Resins filled with fillers based in olive stones present higher modulus than neat polymers, but the strength is affected by the filler content. This drawback does not provide applications for structural applications, but it is a promising solution for non-structural applications. For this purpose, studies covering the fracture and fatigue behavior are expected due to their absence in the literature;-The developed studies reveal that the mechanical properties depend on the filler/matrix interface. The functionalization of the olive stone particles with functionalization or coupling agents showed significant improvements, however, for each type of polymeric matrix, the functionalization agent must be different due to its distinct polarity and physical-chemical interactions. In this context, a complete study on the best functionalization agent and an optimized experimental procedure is highly recommended and should be a priority in future studies;-Olive stone residues constitute a cheap and useful source for reinforcing composites, however, several challenges must still be overcome. Therefore, more research is needed to achieve structural properties and, at same time, to valorize the olive oil industry.

## Figures and Tables

**Figure 1 materials-14-00845-f001:**
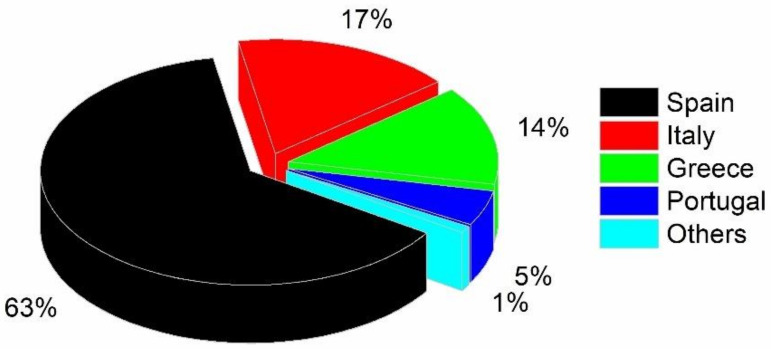
Production percentage of olive oil in the EU: From 2015/16 to 2017/18, average values.

**Figure 2 materials-14-00845-f002:**
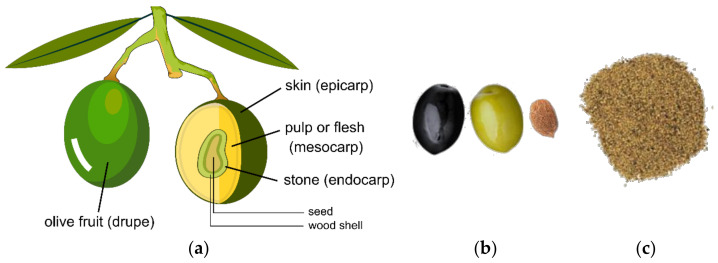
(**a**) Schematic of the cross-section of an olive fruit; (**b**) images of olive fruits and stone; (**c**) image of olive stone granules mechanically grounded, sifted, de-oiled, and crushed from *Olea europaea* L. trees.

**Figure 3 materials-14-00845-f003:**
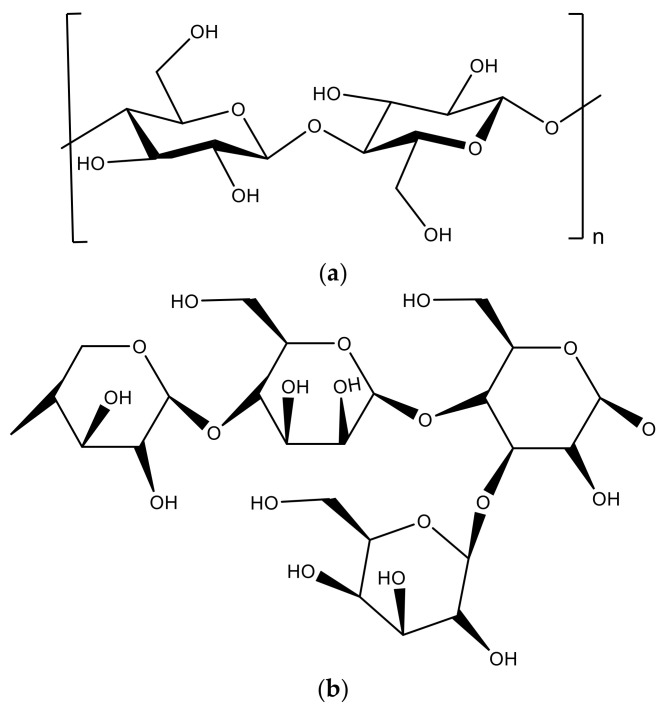
Chemical structures of biopolymers: (**a**) Cellulose, (**b**) hemicellulose, and (**c**) lignin; and monolignols: (**d**) P-coumaryl, (**e**) coniferyl, and (**f**) sinapyl alcohols.

**Figure 4 materials-14-00845-f004:**
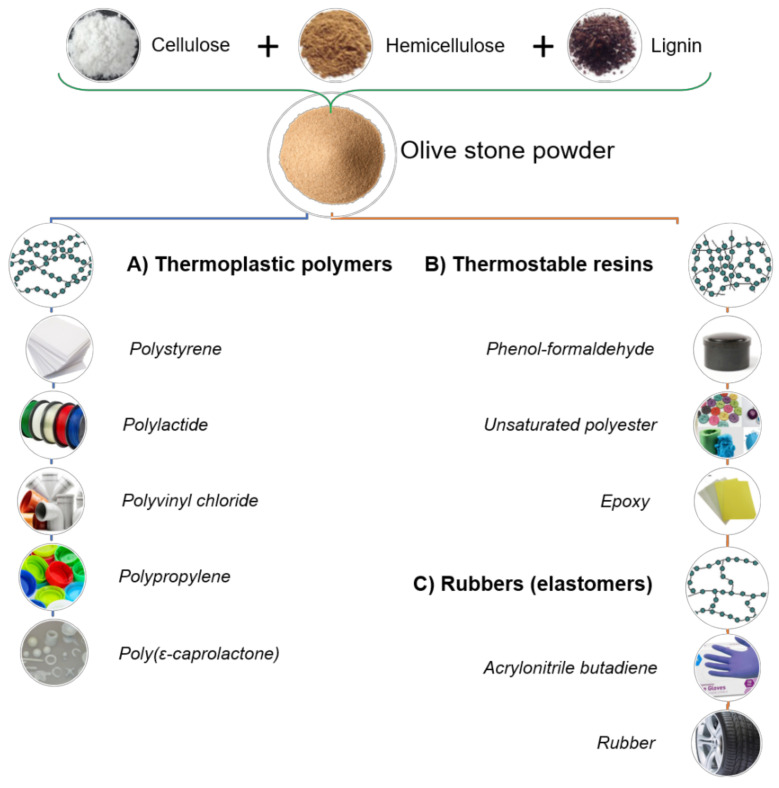
Scheme of the main chemical components present in olive stone powder and main polymer-based composites with olive stones, classified by their crosslinking nature: (**A**) thermoplastic (no cross-linking), (**B**) thermosetting (high degree of cross-linking density), and (**C**) rubbers or elastomers (low cross-link density).

**Figure 5 materials-14-00845-f005:**
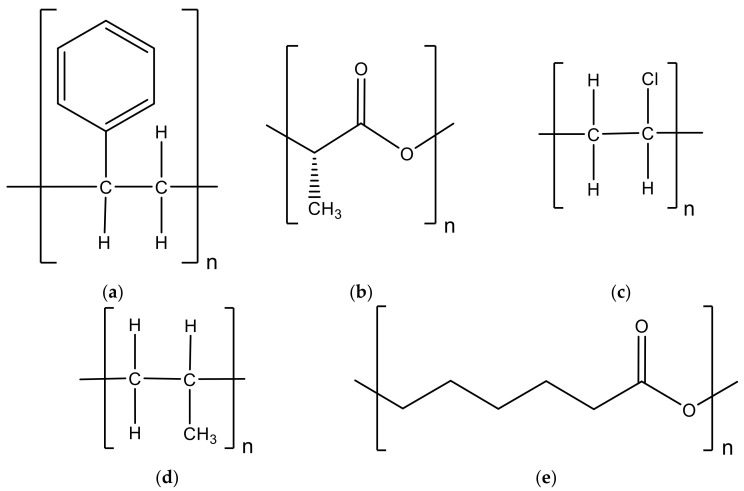
Chemical structures of: (**a**) Polystyrene (PS), (**b**) poly(L-lactide) (PLLA), (**c**) polyvinyl chloride (PVC), (**d**) polypropylene (PP), and (**e**) poly(ε-caprolactone) (PCL) (*n*: Number of repeat units).

**Figure 6 materials-14-00845-f006:**
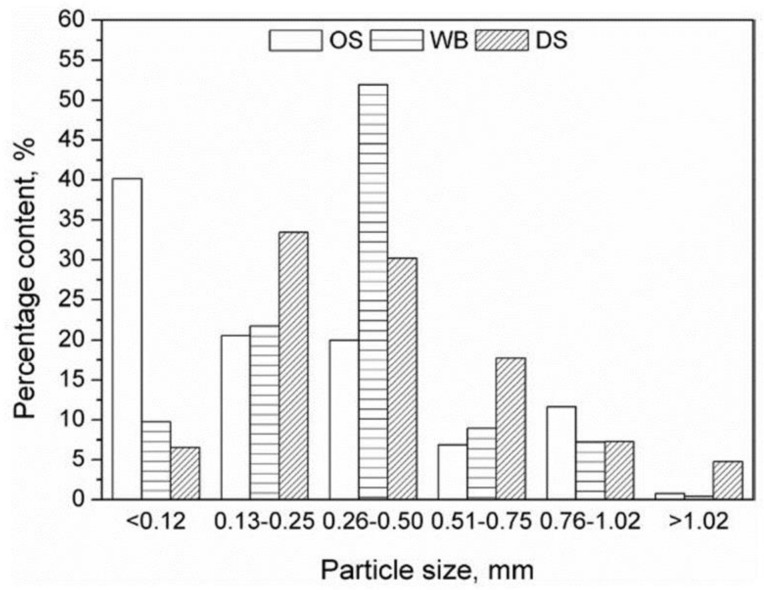
Particle size distribution of analyzed lignocellulosic fillers [52].

**Figure 7 materials-14-00845-f007:**
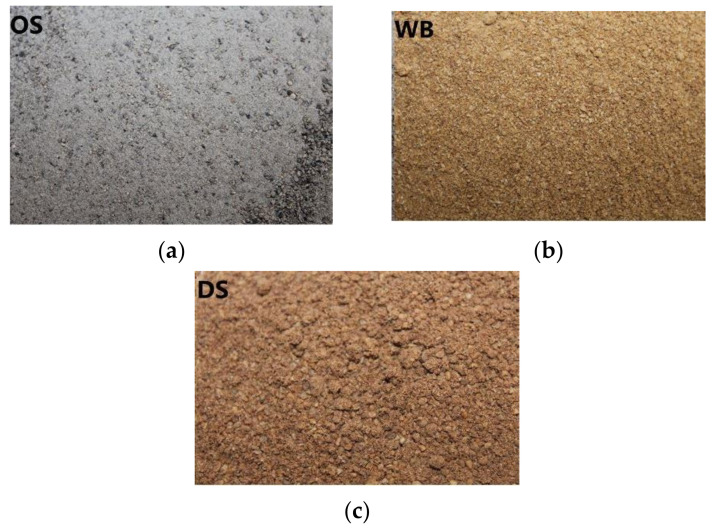
Macroscopic appearance of used lignocellulosic fillers: (**a**) Olive stones (OS), (**b**) wheat bran (WB), and (**c**) date seed (DS) [52].

**Figure 8 materials-14-00845-f008:**
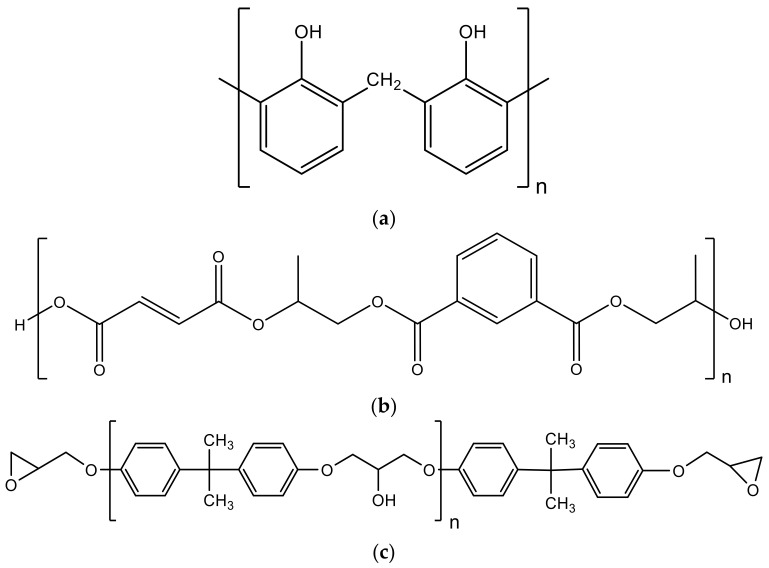
Examples of the chemical structures of: (**a**) Phenol-formaldehyde, (**b**) unsaturated polyester, and (**c**) epoxy resins (n: Number of repeat units).

**Figure 9 materials-14-00845-f009:**
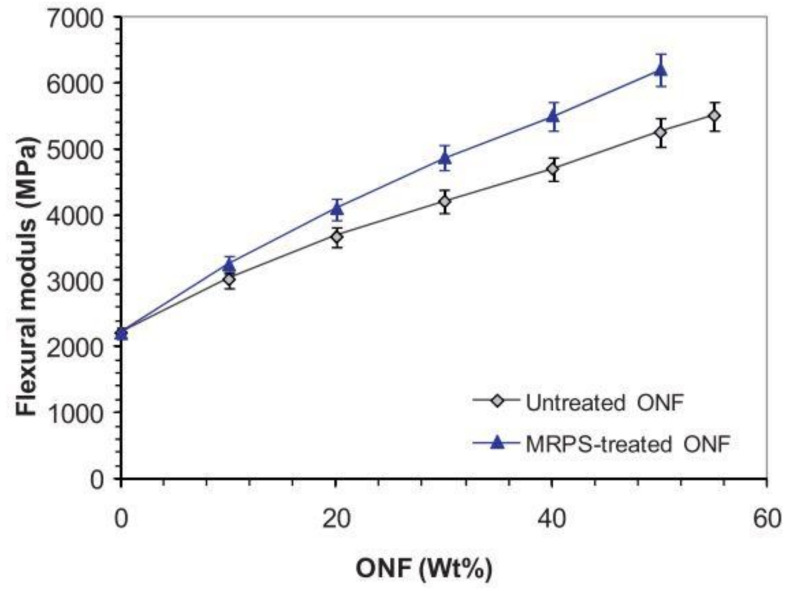
Evolution of the flexural modulus versus ONF (olive nuts flour) content [61].

**Figure 10 materials-14-00845-f010:**
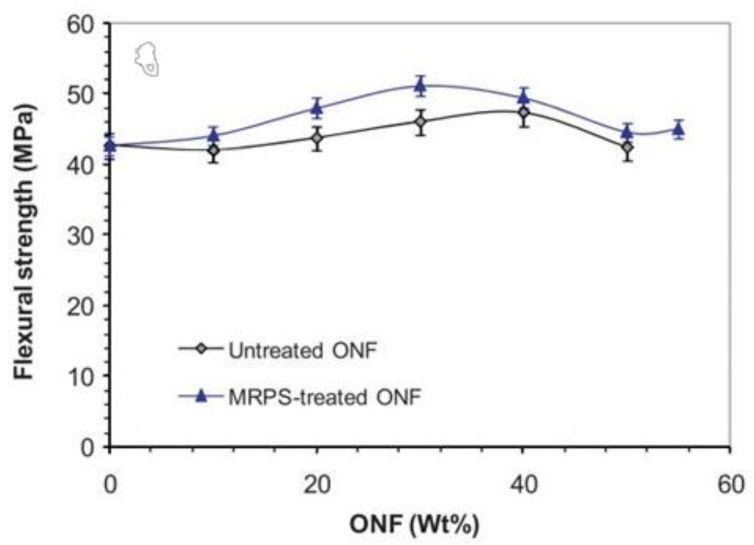
Evolution of the flexural strength versus ONF content [61].

**Figure 11 materials-14-00845-f011:**
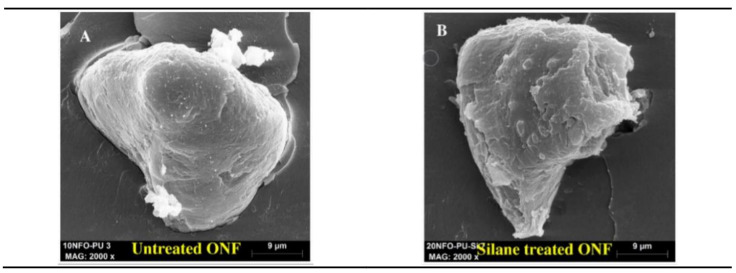
SEM micrograph: (**A**) In the absence; (**B**) in existence of treatment [61].

**Figure 12 materials-14-00845-f012:**
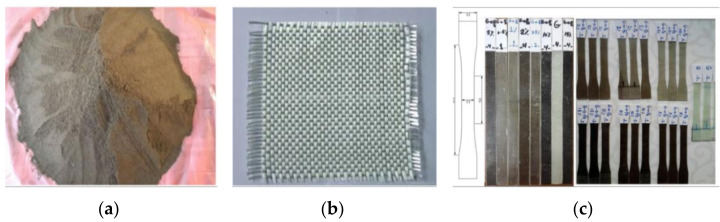
Materials used in production of test samples. (**a**) OP filler, (**b**) S-glass fibers, and (**c**) test samples for dynamic and mechanical tests [65].

**Figure 13 materials-14-00845-f013:**
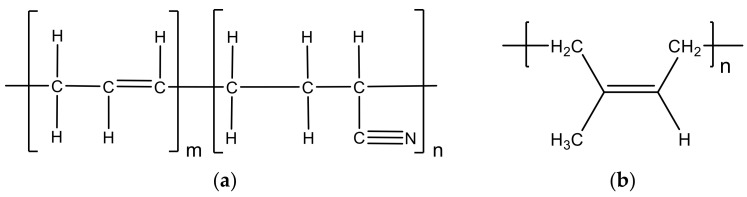
Chemical structures of: (**a**) Acrylonitrile butadiene rubber, and (**b**) natural rubber or poly-cis-isoprene (n, m: Number of repeat units).

**Figure 14 materials-14-00845-f014:**
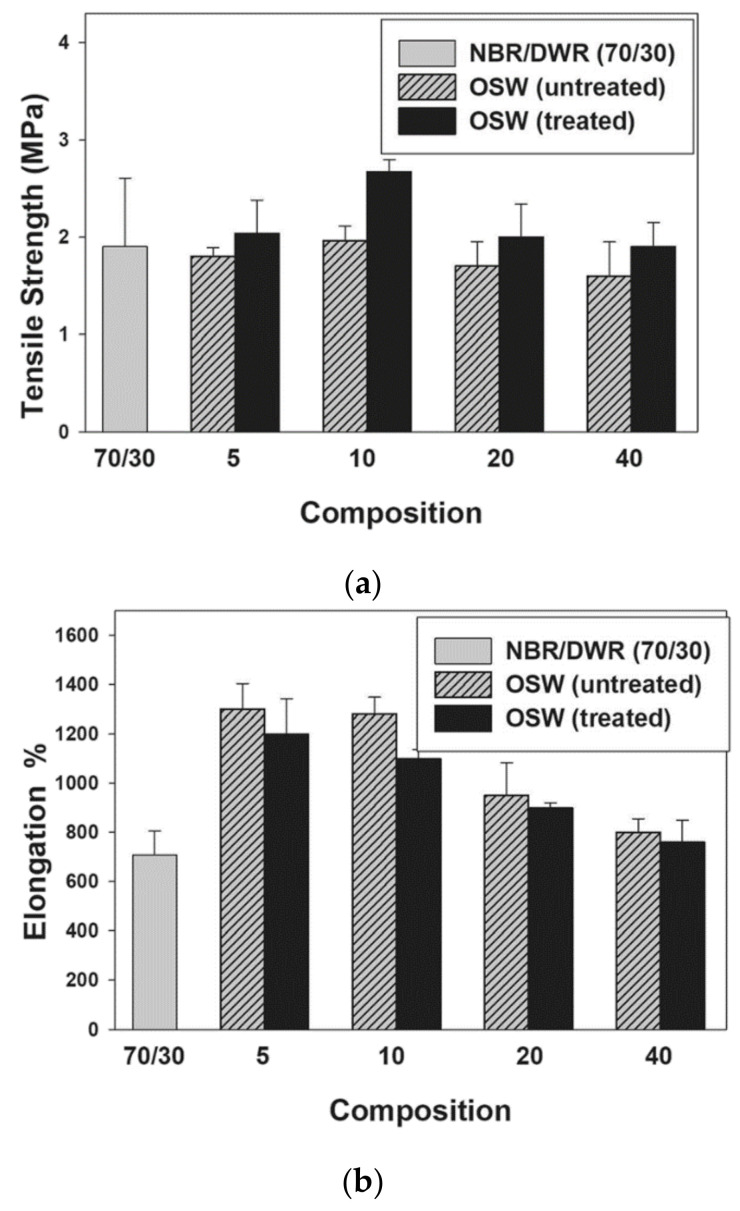
Tensile properties of acrylonitrile butadiene rubber/devulcanized waste rubber (NBR/DWR) blend (70/30) reinforced with different concentrations of (treated or untreated) olive stones waste; (**a**) Tensile Strength (TS), (**b**) Elongation at break % (Eb), (**c**) Elastic Modulus (EM) [68].

**Figure 15 materials-14-00845-f015:**
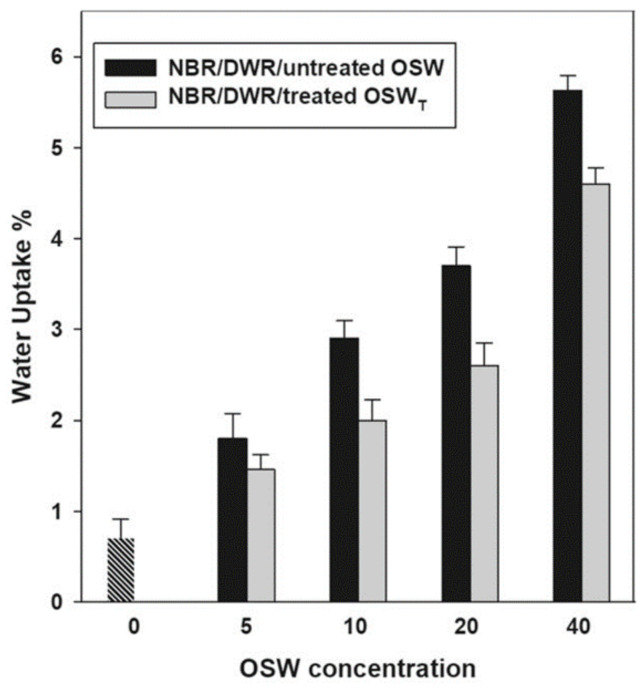
Water uptake % of NBR/DWR blend (70/30) reinforced with different concentrations of (treated or untreated) olive stones waste [68].

**Figure 16 materials-14-00845-f016:**
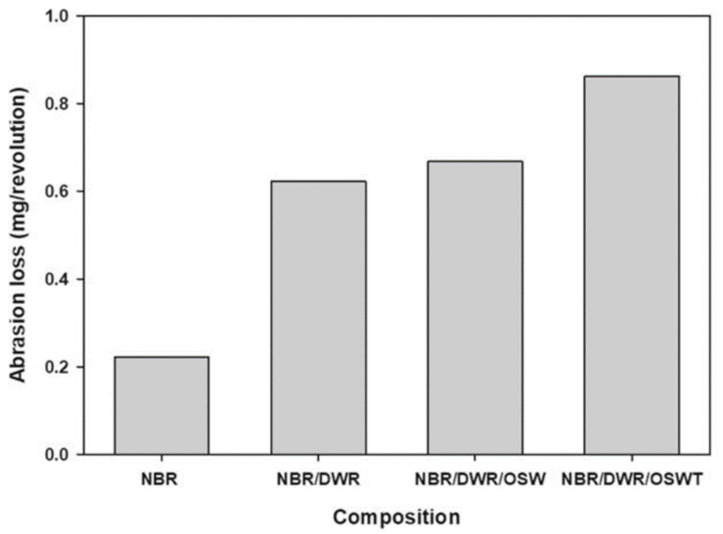
Abrasion resistance of NBR and NBR/DWR (70/30) and NBR/DWR reinforced with treated or untreated olive stones waste [68].

**Table 1 materials-14-00845-t001:** Current and future perspectives for olive stone applications (adapted from [27]).

Application	Raw Transformed	Application Sector
Combustion	Electric or heat	All industries residential and commercial
Activated Carbon	Activated carbon	Food, chemical, petroleum, nuclear, mining, pharmacological industry
Bio-oil	Liquid and gas production	Wide field of industries
Olive seed oil	Olive seed oil	Food, pharmacological, and cosmetic industry
Furfural	Furfural	Wide field of industries as solvent
Plastic filled	Composite	Different industrial applications
Abrasive	Powder	Cleaning
Cosmetic	Cosmetic products	Cosmetic
Biosorbent	Granulated or powder stone	Metallurgy and food
Animal feed	Animal food	Food
Resins	Phenol–formaldehyde	Electrochemical
Fractionation	Soluble phenols and hemicellulose, lignin, and cellulose	Food, cosmetic, pharmaceutical, alcohol

**Table 2 materials-14-00845-t002:** General chemical composition (as % of dry weight), and composition (%) of CHCl_3_–EtOH extractable compounds of olive stones (adapted from [27,29]).

General Chemical Compounds	Percentage (%)	CHCl_3_–EtOH Extractable Compounds	Percentage (%)
Cellulose	31.29	Alkanes	1.7
Hemicellulose	21.9	Triacylglycerols	78
Lignin	26.5	Free fatty acids	7
Moisture	9.79	Aliphatics alcohols	0.1
Fat	5.53	Triterpene alcohols	1.5
Proteins	3.20	Triterpene acids	0.6
Free sugars	0.48	Free sterols	traces
Others	1.31	Steryl esters	1.1
		Unidentified	10

**Table 3 materials-14-00845-t003:** Type of polymer matrix, polymers, properties, characterization, and potential applications or objectives of the works.

Type of Matrix	Polymer	Properties/Characterization	Application/Objectives	Ref.
Thermoplastic polymer	Polystyrene (PS)	Tensile strength, elongation at break, hardness	Manufacture composites with biodegradable properties, light weight, less expensive resources, easy processing, high specific modulus, and environmentally friendly.	[7]
	Recycled post-consumer plastic material	Viscosity, tensile properties (elastic modulus, tensile strength and elongation at break), impact strength	Use very cheap filler in composite manufacture	[8]
	Polylactide (PLA)	Physical, thermal, mechanical properties, tensile modulus		[29]
		differential scanning calorimetry (DSC), thermogravimetric analysis (TGA)		[27]
			Filler	[30]
	Polyvinyl chloride (PVC)	TGA	In plastic-based materials	[31]
	Polypropylene (PP)		In building, automotive industry, and outdoor products, such as deck floors, and furniture, park benches	[32]
		Elasticity modulus		[33,34]
			Industrial applications	[35]
	Polycaprolactone (PCL)	Flexural modulus		[3]
Thermosetting resins	Phenol-formaldehyde	Adsorption	Adsorbents	[36]
				[37]
				[38]
				[39]
				[40]
	Unsaturated polyester	Porosity, particle size, permeability	Particleboards	[4]
			Engineering applications	[5]
	Epoxy	Bending modulus	Adhesive system	[41]
				[42]
			Advanced uses	[43]
				[44]
		Wear rate, hardness		[45]
Rubber or elastomer	NBR/DWR blends	Tensile strength, modulus of elasticity	Rubber industry as lining in fuel tanks and as rubber fuel hoses	[6]

## Data Availability

Data sharing is not applicable to this article.

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
