# Peer review of "Olive Stones as Filler for Polymer-Based Composites: A Review"

_materials, 2021, doi:10.3390/ma14040845_

Round 1

Reviewer 1 Report

Dear Authors,

           Please find an enclosed review of the manuscript (Ms. Ref. No. materials-1079809) entitled Olive stones as filler for polymer-based composites: A review that has been prepared for possible publication in the Materials journal issued by MDPI. Thank you for having me as a referee for the above-mentioned manuscript.

After studying the article given to me for evaluation, I state the following:

In the beginning, I would like to express words of my appreciation for the idea and effort put into conducting research and writing the manuscript recommended to me for review. The authors present polymer composites based on olive stone fillers and different types of polymers: thermoplastic polymers, rubber materials and thermostable resins.

The topic introduces a bit of novelty and is well prepared summarizing findings from other reviews listed in the references. Therefore, the manuscript meets the criteria for publication in the Materials journal issued by MDPI.

However, after reading the manuscript, I have concerns regarding the issues listed below. I would like to ask you to elucidate the vagueness that emerged while reviewing the manuscript so you would be able to improve your investigation.

  1. The authors describe olive stone fillers as an environmentally friendly material but do not include consideration of ecological impact. In the review article, this section should be included.
  2. The synthetic summary should be rebuilt into industrial applications. This section should describe in a remarkably good way where applications are possible and what benefit is derived from them.
  3. Rubbers or elastomeric polymers, please give more examples of the use of olive fillers in this section.
  4. In section 2.2, please list in the table the compounds that are found in the olive stone filler. Very often researchers look for such data and it is very difficult to find them in one place.
  5. The review should include a section related to chitosan currently in use around the world in investigations.
  6. Please improve the style of the English language because there are moments in the text that is not understandable at all.
  7. I think the authors can easily improve it without additional assistance from elsewhere.

Good luck!

Reviewer 2 Report

The paper presents a series of information on use olive stones as filler for polymer-based composites.

From the analysis of the information presented in the article, I found the following:

- The paper presents a series of results that may be of interest to the scientific community;

- the introduction part needs to be substantially improved. In this sense, the part of history must be reduced and more emphasis must be placed on types of composite materials that have organic materials as filling;

- the research methodology should be presented in more detail as it is not very clear;

- a series of composite materials are analyzed that use olive stones as filling material, but for each composite material the same types of properties are not analyzed. I think that the same properties should be analyzed for all composite materials that have different matrix structures and olive stones filling;

- more emphasis must be placed on a graphical analysis of the properties of composite materials with olive stone filling;

- there are paragraphs where the composites are analyzed considering only a scientific paper in the field, which is not enough;

- there are many abbreviations for which there is no specified meaning;

- the percentage of olive stones filling must be completed for all composite materials because there are situations in which this information is missing;

- the discussions should be more applied depending on the results obtained in the analysis of the possibilities of using olive stones for different composite materials.

- discussions should be separated from conclusions;

- in the final part of the conclusions, the future research directions must be presented.

Reviewer 3 Report

The review manuscript entitled: “Olive stones as filler for polymer-based composites:  A review” could be regarded as a very comprehensive review paper giving full insights on the real and possible applications of olive stone waste products as useful organic fillers for wide range of polymer-based composites. Some useful mentioned comparisons with other organic/inorganic reinforcing fillers are also presented and discussed in the review. The combination of three main polymeric/high molecular weight components of olive stones filler material, e.g. cellulose, hemicellulose and lignin is well defended for the complex beneficiary of this filler material as perspective polymer composites additive (both thermoplastics, rubbers (elastomers) and thermoset resins. At the end of the review a very useful Synthetic summery section is given in Table form.

A very few correction recommendations as follow:

Line 144: Lignin is a natural and amorphous biopolymer (biopolymer) constituted by heterogeneous phenyl propane units. Please correct the repeat biopolymer.

Thermo stable resins – please specify the accepted term thermosets as most of the thermostable resins are actually 3D cross-linked one and thus non meltable/thermally stable.

Round 2

Reviewer 1 Report

Accept

Reviewer 2 Report

The authors revised their manuscript according to my suggestions. Thus the manuscript can be accepted for publication.